# The Expression of Allele Changes in NLRP3 (rs35829419) and IL-1β (+3954) Gene Polymorphisms in Periodontitis and Coronary Artery Disease

**DOI:** 10.3390/ma14175103

**Published:** 2021-09-06

**Authors:** Jaideep Mahendra, Abirami Nayaki Rao, Little Mahendra, Mohammed E. Sayed, Maryam H. Mugri, Thodur Madapusi Balaji, Saranya Varadarajan, Raghunathan Jagannathan, Sruthi Srinivasan, Hosam Ali Baeshen, Reji Mathew, Shankargouda Patil

**Affiliations:** 1Department of Periodontology, Meenakshi Ammal Dental College and Hospital, Chennai 600095, India; abirami.nayaki1791@gmail.com (A.N.R.); drsruthisvasan@gmail.com (S.S.); 2Department of Periodontology, Maktoum Bin Hamdan Dental University College, Dubai 213620, United Arab Emirates; littlemahendra24@gmail.com; 3Department of Prosthetic Dental Sciences, College of Dentistry, Jazan University, Jazan 45142, Saudi Arabia; drsayed203@gmail.com; 4Department of Maxillofacial Surgery and Diagnostic Sciences, College of Dentistry, Jazan University, Jazan 45142, Saudi Arabia; dr.mugri@gmail.com; 5Department of Dentistry, Bharathiraja Hospital and Research Institute, Chennai 600017, India; tmbala81@gmail.com; 6Department of Oral Pathology and Microbiology, Sri Venkateswara Dental College and Hospital, Chennai 600130, India; vsaranya87@gmail.com; 7Department of Periodontology, Tagore Dental College and Hospital, Chennai 600127, India; doctorraghunathan@gmail.com; 8Department of Orthodontics, College of Dentistry, King Abdulaziz University, Jeddah 21589, Saudi Arabia; drbaeshen@me.com; 9College of Dental Medicine, Midwestern University, Downers Grove, IL 60515, USA; rmathe@midwestern.edu; 10Department of Maxillofacial Surgery and Diagnostic Sciences, Division of Oral Pathology, College of Dentistry, Jazan University, Jazan 45142, Saudi Arabia

**Keywords:** cardiovascular disease, genetic polymorphisms, inflammasomes, periodontal medicine

## Abstract

Background: Inflammasomes have been shown to play a pivotal role in periodontal disease pathogenesis. However, their role in periodontitis subjects with coronary heart disease remains unclear. This study aimed to obtain the expression of NLRP3 (rs35829419) and IL-1β (+3954) gene polymorphisms in the subgingival plaque and blood samples of generalized periodontitis (GP) subjects with and without coronary heart disease (CHD). Methods: A total of 70 subjects were grouped into two; GP and GP with CHD. Demographic variables and periodontal and cardiac parameters were recorded from both the groups. Subgingival plaque and blood samples were obtained from both the groups and were further subjected to the identification of NLRP3 (rs35829419) and IL-1β (+3954) expression and allele change using a conventional polymerase chain reaction (PCR) and gene sequencing (Sanger’s method). Results: Amongst the demographic variables, age and monthly income were statistically significant between the two groups. Plaque index (PI), clinical attachment level (CAL), high-density lipoprotein (HDL), and low density-lipoprotein (LDL) exhibited statistically significant levels between the two groups. The NLRP3 (rs35829419) and IL-1β (+3954) genes showed a statistically significant association with allele change (frequency) among the groups. The general comparison of all the parameters with the allele change of NLRP3 (rs35829419) and IL-1β (+3954) in the subgingival plaque and blood samples showed statistically significant associations among the two groups. Conclusion: The present study highlighted an allele change in IL-1β (+3954) gene polymorphisms which may play an important role in the pathogenesis of periodontitis and coronary heart disease.

## 1. Introduction

Periodontitis is a chronic, recurrent inflammatory disease caused by bacteria that destroy the tooth’s supporting structure. Since ancient times, it has been one of the most common diseases afflicting mankind. It is caused by a mismatch between the number of pathogenic microbes in the subgingival environment and the host’s immune response. Several risk factors, both modifiable and non-modifiable, may alter the immune response, including genetic risk factors which fall under the non-modifiable category. Furthermore, genetic factors influence the host’s response to a bacterial insult and take into account disease distribution, severity, and spread variation [1].

In recent years, research into the host’s immune response has offered useful insights into the pathogenesis of chronic periodontitis. It has been shown that a dysbiotic bacterial culture is to blame for the periodontal tissue damage by perpetuating inflammation in the soft tissues surrounding the tooth [2]. This dysbiotic flora also regulates the host’s immune responses, which are heavily affected by the individual’s genotype. As a result, while dysbiosis does not cause periodontitis on its own, it may do so when combined with various risk factors including genetic constitution, age, and behavioral activities such as smoking.

The activation of the Toll-like receptors (TLR) and NOD-like receptor pyrin domain containing three (NLRP3)/apoptosis-associated speck-like proteins with a Caspase recruitment domain (ASC)–Caspase-1 multiprotein complex termed as the NLRP3 inflammasome has been studied [3,4]. The formation of the inflammasome results in the auto-proteolytic maturation of caspase-1, which results in the maturation and extracellular release of the pro-inflammatory cytokines interleukin (IL)-1 and IL-18, which have been linked to inflammatory periodontal tissue destruction, possibly via pyroptosis, which is a variety of apoptosis induced by pathogenic bacteria [5].

The upregulation of the release of prostaglandin E2 (PGE2), T-lymphocyte stimulation and cytokine formation, B-lymphocyte proliferation and antibody production, and fibroblast proliferation are all known biological effects of IL-1 [6]. IL-1 also influences neutrophil chemotaxis and activation, as well as endothelial cell activity, and endorses osteoclast formation and bone resorption [7]. Periodontal disease has been linked to a number of systemic conditions, including cardiovascular disease, metabolic syndrome, osteoporosis, and preterm low birth weight. Shared risk factors, similar roles of monocytes, increased fibrinogen and WBC count, the impact of bacterial LPS, oral microbes, and the involvement of the C-reactive protein as a strong biomarker are all possible linking mechanisms that are involved between periodontal disease and coronary heart disease.

Growing evidence indicates that genetic mutations of the NLRP3 gene cause the NLRP3 protein to be constitutively activated, resulting in an unregulated IL-1β output [8]. Alternatively, single-nucleotide polymorphisms (SNPs) in the NLRP3 gene located on chromosome 1q44 have been proposed as a way to influence NLRP3 transcriptional behavior. Nonetheless, the results of the IL-1 (+3954) SNP’s association with chronic periodontitis and coronary heart disease have been contradicting with those of some previous studies, thereby no evidence of a link between the IL-1 +3954 SNP and GP [9] has been elicited, whereas others have found a clear link between the CT genotype and disease severity in various populations [10,11]. While studies have been performed to identify these polymorphisms individually in each disease, none of the studies have been completed so far that have quantified and correlated NLRP3 and IL-1β SNPs in subjects with chronic periodontitis and coronary artery disease. As a result, this study aimed to see if there was a connection between the NLRP3 (rs35829419) and IL-1β (+3954) SNPs and the etiopathogenesis of GP and GP + CHD.

## 2. Materials and Methods

### 2.1. Design of the Study

A total of 110 subjects (male subjects) aged between 30 and 65 years from the Department of Periodontology, Meenakshi Ammal Dental College and Hospital, Chennai, and Frontier Lifeline Hospital, Mogappair, Chennai were screened for the study during the period from December 2018 to November 2020. Of these, 22 individuals refused to take part in the study, and 18 were excluded due to systemic disease involvement such as diabetes mellitus and anemic conditions which were revealed after blood investigation (Figure 1).

Finally, based on the inclusion criteria, 70 subjects were grouped into 2; the GP Group was comprised of chronic periodontitis subjects without coronary heart disease and the GP + CHD group was comprised of chronic periodontitis subjects with coronary heart disease (Figure 1). The power analysis of the study was determined to be 80% with a minimum sample size of 70 samples based on the prevalence of periodontitis and coronary heart disease taken from hospital records and previous evidence from the literature. The study was approved by the “Institutional Review Board”, MAHER-Deemed to be University, Chennai (MADC/IRB-XXV/2018/391 A). The study was conducted under the Helsinki Declaration of 1975, as revised in 2013. Written informed consent was attained before the start of the study after the study protocol was explained to all the individuals.

### 2.2. Inclusion and Exclusion Criteria

The inclusion criteria for the selected subjects were as follows: (1) patients who were willing to take part in the study; (2) patients who fell into the age group of 30–65 years (male); (3) patients with at least 20 remaining natural teeth; (4) in the GP group, individuals with periodontitis who satisfied the case definition by the World Workshop in the Classification of Periodontal Diseases and Conditions 2017 (Stage 2, Grade B periodontitis), were recruited; (5) for both the GP and GP + CHD group, the patients with cardiac problems were diagnosed based on a clinical examination, whether the patient was symptomatic or asymptomatic, an ECG, and a blood test for the lipid profile and LDH followed by an echocardiogram. Based on the above diagnostic parameters, the appropriate diagnosis was made by the cardiologist for the patients. The exclusion criteria for both groups included: (1) systemic conditions such as type I and type II diabetes mellitus, renal disease, rheumatoid arthritis, respiratory diseases, allergy, liver disease, HIV infection, and advanced malignancies; (2) female patients were excluded due to hormonal variations; (3) the use of corticosteroids antibiotics within 3 months of the investigation; (4) current smokers and individuals who had quit smoking less than 6 months ago; (5) periodontal therapy within the previous 6 months.

### 2.3. Demographic, Clinical, and Cardiac Parameters

All the participants underwent documentation of their demographic variables such as age, height, weight, BMI, waist–hip ratio, and income per month. Periodontal parameters such as the plaque index (PI) [12] was measured at 4 sites of all the teeth of the dentition, bleeding on probing (BOP) was measured in 4 sites of all the teeth of the dentition and expressed in % [13], the probing pocket depth (PPD) (mm) in 6 sites of each tooth in the entire dentition was measured and averaged, and the clinical attachment level (CAL) (mm) was measured in 6 sites per tooth in the entire dentition and averaged. Cardiac parameters such as the total cholesterol levels (TC), high-density lipoprotein levels (HDL), low-density lipoprotein levels (LDL), total triglyceride levels (TG), systolic blood pressure (SBP) and systolic blood pressure (DBP) were recorded. The periodontal parameters were recorded using a Williams periodontal probe (Manual Williams periodontal probe, Hu-Friedy, Chicago, IL, USA) to the nearest millimeter from six sites per tooth.

### 2.4. Subgingival Plaque and Blood Samples Collection

After periodontal examination and before the periodontal treatment period, subgingival plaque samples were obtained from the deepest periodontal pocket with the help of a Universal curettes (Manual Universal Curettes, Hu-Friedy, Chicago, IL, USA) from subjects in both the groups and were transferred to the sterilized vial containing 95% absolute ethanol and stored at −80 °C until further analysis. Following the subgingival plaque collection, 2 mL of venous blood was collected from the antecubital vein by venipuncture using a 20-gauge needle with a 2 mL syringe from each participant in both groups. The collected blood sample was immediately transferred to the EDTA vacutainer and stored at 4 °C for further analysis.

### 2.5. PCR Analysis

A total of 100 µL of DNA was isolated from the subgingival plaque samples using an Xpress DNA kit (Xpress DNA kit, MagGenome Technologies Pvt. Ltd., Cochin, India) and blood samples using the QIAamp DNA mini kit (QIAamp1 DNA mini kit, Qiagen Sciences, Germantown, MD, USA), respectively, as per the manufacturer’s protocol. After the isolation of the DNA, quantification was performed using a nanodrop machine (NANODROP, Thermo Scientifics, Waltham, MA, USA) and a PCR (PCR, Agilent Technologies, SureCycler 8000) analysis was processed. Specific primers (Oligonucleotide primer, PrimeX) were used to determine the inflammasomes NLRP3 (rs35829419) (Forward primer: CAGACTTCTGTGTGTGGGACTGA and Reverse primer: TCCTGACAACATGCTGATGTGA) and IL-1 (+3954) (Forward primer: GGCCTGCCCTTCTGATTTTATA and Reverse primer: TCGTGCACATAAGCCTCGTTA).

The reaction mix (20 μL) was prepared by adding 10 μL of the PCR master mix solution, 2 μL of the forward primer (Oligonucleotide primer, PrimeX), 2 µL of the reverse primer (Oligonucleotide primer, PrimeX), and 2.5 µL of DNA, and the solution was made up to 20 µL with sterile water. The reaction conditions for both the primers were as follows: initial denaturation at 95 °C for 10 min, followed by 40 cycles of PCR with denaturation at 95 °C for 15 s, annealing at 53 °C for 60 s for IL-1β, and annealing at 60 °C for 60 s for NLRP3 followed by an extension at 72 °C for 60 s for both the primers. The final extension was completed at 72 °C for 60 s.

### 2.6. Agarose Gel Electrophoresis

A total of 1 g of agarose was dissolved in 50 mL of a Tris-acetate EDTA 1X (TAE) buffer and heated. Then, the agarose gel was cast with 5 µL of ethidium bromide.

The solution was then poured into the gel cassettes and allowed to cool and solidify. Then, the gel cassette was placed in the tank containing the 50X TAE buffer. A total of 5 µL of the PCR product was then loaded to each well with a 100 bp DNA ladder to track the molecular weight of the PCR product. Then, the electrophoresis was carried out at 150 mA and the gels were visualized in a Gel Doc machine.

### 2.7. Gene Sequencing

For the Sanger sequencing, a complementary DNA primer to the template DNA (the DNA to be sequenced) was used as a starting point for DNA synthesis. In the presence of the four deoxynucleotide triphosphates (dNTPs: A, G, C, and T), the polymerase extended the primer by the addition of the complementary dNTP to the template DNA strand. The ends of the fragments were then labeled with dyes that indicated their final nucleotide.

## 3. Statistical Analysis

Statistical analysis was completed using the Statistical Package for Social Sciences (SPSS) version 20.1 (Chicago, USA Inc). For both groups, the mean and standard deviation for all parameters were calculated. The independent sample t-test was used to compare all variables between the groups. The Chi-square test was used to determine the relationship between the categorical variables. The one-way analysis of variance (ANOVA) test was used to compare the categorical and quantitative variables. In the current study, p 0.05 was considered significant.

## 4. Results

### 4.1. Demographic Variables, Periodontal, and Cardiac Parameters

The results of the bivariate comparison among the demographic variables and the clinical parameters, the mean age (*p*-value < 0.001), monthly income (*p*-value 0.034), the plaque index (*p*-value < 0.001), the clinical attachment level (*p*-value < 0.001), the high-density lipoprotein (*p*-value 0.019), and the low-density lipoprotein (*p*-value 0.006) exhibited statistically significant levels between the two groups (Table 1). Other demographic variables (height, weight, BMI, waist–hip ratio), periodontal (BOP, PPD), and the cardiac parameters (TC, TG, SBP, DBP) did not show any statistical difference (Table 1).

### 4.2. PCR Analysis

The n (%) and mean of the NLRP3 (rs35829419) and IL-1β (+3954) gene polymorphisms in the subgingival plaque and blood samples were found to be higher in the GP + CHD group compared to the GP group (Table 2). However, both the inflammasomes were statistically significant in the subgingival plaque and blood samples (Table 2).

### 4.3. Gene Sequencing Analysis

The IL-1β (+3954) gene showed a statistically significant association with CT (*p*-value 0.052), AT (*p*-value 0.01), and AG (*p*-value 0.032) allele changes (frequency) in the subgingival plaque samples. Similarly, a statistically significant association between CT (*p*-value 0.026) and AT (*p*-value 0.007) was observed in IL-1β (+3954) and a non-significant association between AG (*p*-value 0.78) and allele changes (frequency) was perceived in the blood samples in both the groups (Table 3). NLRP3 (rs35829419) was not selected for sequencing as the PCR band strength was not high and its intensity was low.

From a comparison of the demographic, periodontal, and cardiac parameters with the allele change (frequency) of IL-1β (+3954) in both the subgingival plaque and blood samples, the results showed a statistically significant association with age (*p*-value 0.001), CAL (*p*-value 0.01), diastolic blood pressure, and bleeding on probing. IL-1β (+3954) in the blood samples also showed a statistically significant association with monthly income (*p*-value 0.01) and plaque index (*p*-value 0.05) (Table 4).

## 5. Discussion

The oral cavity is home to a complex environment with a wide range of microorganisms. It acts as a reservoir or niche for bacteria that can spread systemically and cause a systemic inflammatory response. The pathogenic bacteria circulate continuously in chronic periodontitis, raising the inflammatory response and the activation of the host’s immune system, resulting in unpredictable inflammation and extreme damage to the alveolar bone and the soft tissues surrounding the tooth [14].

The invasion and destruction of the gingival epithelium by periodontal pathogens results in the release of numerous local inflammatory mediators into the periodontal pocket, which then enter the systemic circulation, enabling immune cells to be recruited. Bacteria can travel across the bloodstream in two ways: indirectly and directly [15]. As a result, bacteria components and systemic inflammatory mediators can speed up the formation of coronary plaques in people who are predisposed to cardiovascular disease (CVD) (e.g., genetic, lifestyle). Periodontal bacteria have been found in a variety of tissues and organs, the most prominent of which is the cardiovascular system [16,17].

Inflammasomes are the main regulators of the innate immune system in chronic inflammatory diseases, detecting and fighting microbial pathogens and thereby regulating and reducing the invading microbes. They play a role in the development of periodontitis and the production of inflammasome-related inflammatory mediators [18]. Several forms of NLRs have been linked to systemic diseases (NLRP1, NLRP2, NLRP3, NLRP6, NLRP12). The pathogenesis of many inflammatory disorders, including atherosclerosis, gout, Crohn’s disease, and periodontitis has been related to the inflammasomes NLRP3 and IL-1 [18].

The activation of the NLRP3 inflammasome in response to a variety of stimuli has led to the hypothesis that different agonists trigger similar downstream events that are detected by the NLRP3 inflammasome. Inactive caspase-1 is converted to active caspase-1 when NLRP3 is activated [19]. IL-1β (+3954), on the other hand, is a pro-inflammatory cytokine that plays a vital role in immune control, inflammation, and bone resorption in periodontitis. Patients with periodontitis have substantially higher intensities of IL-1β in salivary and gingival crevicular fluid samples than in healthy controls [1].

The evidence strongly indicates that genetic contributions to inflammatory-induced diseases such as chronic periodontitis and coronary artery disease are perplexing, given factors such as genetic variability, gene-environmental interactions, and inadequate penetrance [20].

This is the first study to evaluate the expression of NLRP3 (rs4612666) and IL-1β (+3954) gene polymorphisms in subgingival plaque and blood samples. The demographic variables such as height, weight, BMI, the waist–hip ratio did not show any statistical difference, whereas age and monthly income displayed statistically significant levels between the two groups (Table 1). Age was found to be significantly higher in the GP + CHD group than in the GP group, indicating that age is a confounding factor for periodontal and coronary heart disease. The literature proposed that previous reports of the National Heart, Lung, and Blood Institute (NHLBI) demonstrated that males, especially those 45 years of age and above have an increased prevalence of coronary heart disease [21]. Hence, in the present study, coronary heart disease subjects with periodontitis showed a higher mean age (>40 years). The mean monthly income in the subjects with GP + CHD was comparatively lower than the GP group which was statistically significant (Table 1). The existing results were in accordance with Janati, A et al. who demonstrated that lower or middle social classes presented a greater risk for coronary heart disease than higher social class individuals [22]. The prevalence of CHD in individuals from a lower social class may suggest that dietary intake and lifestyle are due to low incomes. In developing countries such as India, due to rapid urbanization people are becoming more aware about healthy lifestyles and education which can be attributed to the decrease in CHD among these individuals.

Periodontal parameters such as PI and CAL showed statistically significant levels in the GP + CHD group compared to the GP group with a *p*-value of <0.001, respectively, suggesting a severity of periodontitis in GP + CHD subjects. Even though periodontal parameters measured clinically cannot be a sole indicator of disease severity and the association with CHD remains inconclusive, the link proven from the study by de Macêdo et al. also reported a causal relationship [23]. de Macêdo et al. stated that individuals with a plaque index of ≥65% were positively associated with periodontal destruction which is in concurrence with the findings of the current study [23]. The association between clinical attachment level and cardiovascular diseases has also been explored in the previous literature [24,25]. Moreover, epidemiological data [26,27] revealed an association of periodontal parameters such as BOP, PD, and CAL with the degree and number of obstructed coronary arteries which indicated a positive correlation for periodontal and cardiovascular disease [28]. Thus, in this study, the GP + CHD group had a maximum value of the periodontal parameters showing more periodontal destruction compared to the GP group, which further establishes a strong link between periodontitis and CHD.

Likewise, cardiac parameters such as TC, TG, SBP, and DBP showed a higher mean in the GP + CHD group compared to the GP group but were not statistically significant. In contrast, HDL and LDL revealed statistically significant mean levels in GP + CHD subjects compared to GP subjects with a *p*-value of 0.019 and 0.006, respectively. This is in accordance with the results of Akkaloori, A. et al., who demonstrated that the mean levels of HDL, LDL, and TC were significantly higher in coronary heart disease patients with periodontitis [29]. As antioxidant levels decline in a periodontitis patient, there is a greater likelihood of an increased oxidation of LDL and a decreased HDL, thereby increasing the oxidative state. Under oxidative stress, HDL proteins undergo changes that reduce their atheroprotective properties, influencing artherogenesis [30]. Bengtsson and Karlsson suggested that cardiac parameters play a role in the inflammatory pathway for periodontal and cardiovascular diseases; a possible systemic inflammation caused by periodontal pathogenic bacteria may result in increased LDL and decreased HDL levels in the bloodstream, implying an increased risk of coronary heart disease [30].

Molecular genetic research has recognized the individual genetic makeup that may promote the development of periodontitis and cardiovascular disorders. It has also been speculated that inflammation-related genetic variation increases the susceptibility to periodontitis and coronary heart disease, although this is yet to be established. As a result, we decided to explore the involvement of NLRP3 and IL-1β SNPs in inflammatory protein expression. To the best of our knowledge, no research has looked into the role of NLRP3 and IL-1β single nucleotide polymorphisms in patients with periodontitis and CHD.

The foremost findings reported significantly higher mean levels of NLRP3 (rs35829419) and IL-1β (+3954) gene polymorphisms in the subgingival plaque samples and blood samples of the GP + CHD group compared to the GP group (Table 2). Samples were selected for sequencing based upon the PCR band strength and intensity. A significant deviation from the Hardy–Weinberg equilibrium X^2^ (HWE) denoted a genotyping error and a sampling bias. After analyzing both the SNPs of IL-1 (+3954) in the subgingival plaque and blood samples in the current study, it was discovered that both SNPs were proportionately compatible with the HWE X2 among the GP and GP +CHD subjects, indicating that the frequencies of the occurrence of the alleles in these SNPs in the population were stable.

In our study, we found a significant allele change in the CT value of NLRP3 (rs35829419) in the subgingival and blood samples in both the groups. The GP+CHD group showed a higher odds ratio compared to the GP group alone. This is in accordance with the previous study by Mahendra J et al. who also demonstrated the significant change in NLRP3 (rs4612666) in cardiac patients with generalized periodontitis compared to non-cardiacs [31]. The rise in NLRP3 (rs35829419) polymorphism suggests that it recognizes the bacteria indirectly by monitoring the host-derived DAMP levels that are the product of tissue or cell injury caused by bacterial toxins [18].

In the existing study, the allele change of CT in IL-1β (+3954) in the subgingival plaque and blood samples showed a significant odds ratio among the GP and GP + CHD groups, respectively. Similarly, the allele AT showed a significant odds ratio in the subgingival plaque and blood samples. Moreover, the allele AG obtained a statistically significant odds ratio in the subgingival plaque samples and statistically non-significant odds in the blood samples (Table 3). A former study by Kim et al. showed a significant odds ratio of the CT allele when comparing periodontally diseased subjects with the healthy group. The results suggested that the genetic variations of interleukin -1β are associated with an increased risk of periodontitis [32]. A study by Massamatti et al., observed a nonsignificant odds ratio of 1.3–1.8 of the CT allele change in chronic periodontitis patients when compared with aggressive periodontitis patients [10]. Hence, it can be stated that individuals carrying the rare variant of the IL-1 (+3954) polymorphism would thus have increased inflammatory activity, according to previous research. It is also important to highlight that a positive association of the allele change of IL-1β (+3954) with age, BOP, and CAL was observed in the subgingival plaque and blood samples. In the blood samples alone, significant associations with monthly income and PI were seen (Table 4). This is in accordance with Massamati et al. who also observed a higher CAL and BOP in chronic periodontitis patients with a higher prevalence of IL-1β allele change [10]. Our findings are also in accordance with a study by Hodge PJ et al. and Moreira et al. who observed the association of IL-1B polymorphism with chronic periodontitis in Australian Caucasian patients and the Brazilian population, respectively [33,34]. One plausible mechanism for the association between the IL-1 B gene polymorphism and periodontal disease may be due to the fact that polymorphic genotypes of IL-1 β would directly have an impact on the disease progression by influencing the cytokine synthesis. The IL-1β polymorphism could play a direct role in periodontal inflammation and destruction. In the current study, we did not find any significant difference between the other parameters and the allele change frequencies of IL-1β SNPs which was indicative of factors such as anatomic variations, local irritants, and pathogens which cover the genetic influence of IL-1β SNPs on chronic periodontitis and coronary heart disease development. Previous research has found conflicting evidence linking the NLRP3 and IL-1 polymorphisms to a variety of inflammatory diseases, including inflammatory bowel disease, n increased severity of rheumatoid arthritis, an increased risk of Alzheimer’s disease, and other inflammatory diseases [18].

Despite the controversy associated with research on the role of genetics, the assessment of SNPs’ presumed functionality in the oral cavity may expose their pathogenic role in inflammatory diseases such as periodontitis and coronary heart disease. There has been little research on the correlation between inflammasome SNPs in periodontitis and coronary heart disease. To the best of the authors’ understanding, this is the first research to look at this relationship in depth. Longitudinal studies are required to investigate the function of NLRP3 (rs4612666) and IL-1β (+3954) gene polymorphisms as therapeutic markers for periodontitis and coronary heart disease. Periodontal therapy is moving towards the paradigm of personalized medicine and appropriate genetic testing to identify the defects at the basic level of the genes present in the human body. Hence, the identification of important genetic polymorphisms such as NLRP3 (rs4612666) and IL-1β (+3954) will pave the way for the development of point of care treatment for patients suffering from periodontitis and coronary heart disease. Further, chair-side diagnostic kits can be developed for the early detection of these polymorphisms which can aid in more accurate treatment plans for these patients. 

## 6. Limitations

There were few limitations noted for the study. First, only the CHD patients were selected for the study but the severity of coronary heart disease was not assessed. In the future, the research pertaining to the biomarkers along with the severity of the disease would be more significant if the role of these polymorphisms with CHD was evaluated. Secondly, the sample size was limited. A larger sample size with fewer dropouts would be beneficial for future longitudinal studies. In the present study only the expression of NLRP3 (rs35829419) and IL-1β (+3954) polymorphisms was noted. Thus, further studies are required to evaluate and explore the prognostic and therapeutic implications of these biomarkers in the progression of periodontal and coronary heart disease.

## 7. Conclusions

These findings may serve as a foundation for potential genetic mutation research into inflammatory diseases, such as the relationship between periodontitis and coronary heart disease, based on these facts. An increased knowledge of gene polymorphisms may lead to new therapeutic approaches for periodontal disease, atherosclerosis, and other inflammatory diseases.

## Figures and Tables

**Figure 1 materials-14-05103-f001:**
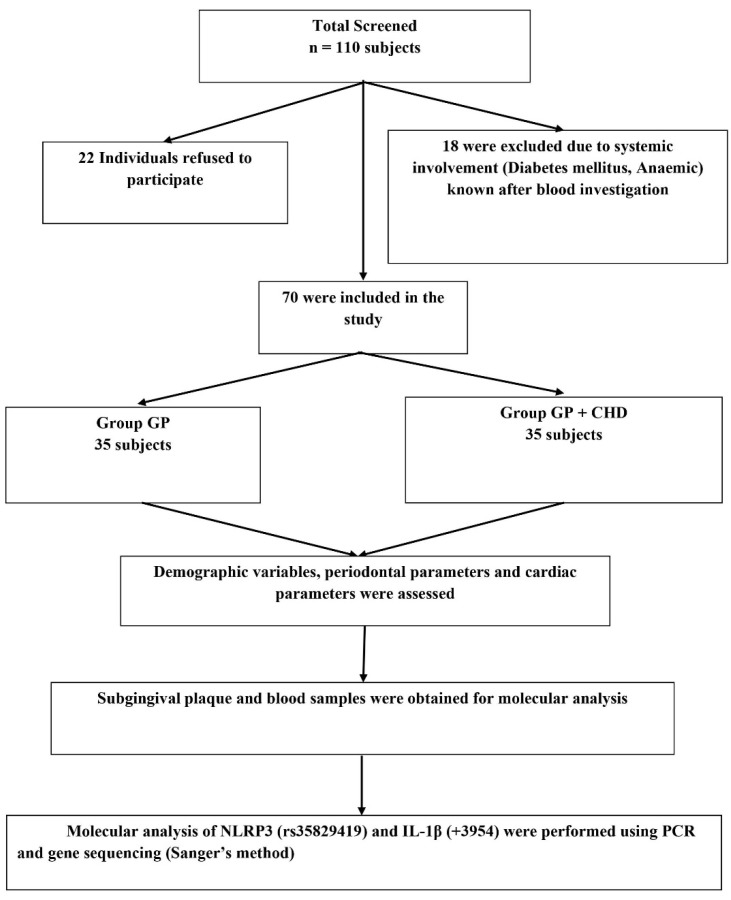
Summary of the study methodology.

**Table 1 materials-14-05103-t001:** Comparison of the demographic variables, the periodontal parameters, and the cardiac parameters between Group GP and Group GP + CHD.

Variable	Group GP Mean ± Standard Deviation	Group GP + CHD Mean ± Standard Deviation	*p*-Value
Age (years)	43.42 ± 9.08	56.45 ± 8.32	<0.001^‡^
Height (cm)	166.00 ± 6.11	164.08 ± 9.43	0.317^NS^
Weight (Kg)	65.40 ± 6.40	67.50 ± 10.43	0.312^NS^
Body Mass Index (BMI)	23.65 ± 2.49	24.84 ± 3.14	0.083^NS^
Waist–Hip ratio	0.83 ± 0.04	0.84 ± 0.04	0.474^NS^
Monthly Income (Rs)	23342.86 ± 7870.05	22000 ± 7352.47	0.034^*^
Plaque Index	0.60 ± 0.12	0.95 ± 0.15	<0.001^‡^
Bleeding on Probing (%)	71.22 ± 8.09	76.76 ± 24.31	0.205^NS^
Probing Pocket Depth (mm)	3.71 ± 0.677	18.03 ± 79.51	0.291^NS^
Clinical Attachment Level (mm)	4.48 ± 0.60	5.69 ± 0.55	<0.001^‡^
Total Cholesterol (mg/dl)	173.14 ± 28.91	187.51 ± 46.47	0.125^NS^
High-Density Lipoprotein (mg/dl)	40.60 ± 6.26	36.37 ± 8.28	0.019^†^
Low-Density Lipoprotein (mg/dl)	107.31 ± 23.16	129.11 ± 39.50	0.006^†^
Triglycerides (mg/dl)	133.11 ± 29.98	144.40 ± 55.94	0.297^NS^
Systolic Blood Pressure (mm Hg)	125.14 ± 8.86	125.42 ± 9.50	0.897^NS^
Diastolic Blood Pressure (mm Hg)	78.28 ± 5.13	80.28 ± 6.17	0.145^NS^

* *p* < 0.05 is considered to be statistically significant. † *p* < 0.01 is considered to be highly statistically significant. ‡ *p* < 0.001 is considered to be very highly statistically significant. NS—Nonsignificant.

**Table 2 materials-14-05103-t002:** Comparison of NLRP3 (rs35829419) and IL-1β (+3954) gene expression between Group GP and Group GP + CHD.

Gene Expression	Group GP		Group GP+CHD		*p*-Value
	n (%)	Mean ± Standard Deviation	n (%)	Mean ± Standard Deviation	
**NLRP3 (rs35829419) (Subgingival plaque sample)**	0	0.00 ± 0.00	5.70	0.57 ± 0.23	0.003^‡^
**NLRP3 (rs35829419) (Blood sample)**	11.43	0.11 ± 0.32	34.29	0.34 ± 0.48	0.000^‡^
**IL-1β (+3954) (Subgingival plaque sample)**	82.9	0.82 ± 0.38	91.4	0.91 ± 0.28	0.032^†^
**IL-1β (+3954) (Blood sample)**	71.4	0.71 ± 0.45	100	1.0 ± 0.00	0.000^‡^

† *p* < 0.01 is considered to be highly statistically significant. ‡ *p* < 0.001 is considered to be very highly statistically significant. NS—Nonsignificant.

**Table 3 materials-14-05103-t003:** Allele change (frequency) of IL-1β (+3954) gene polymorphisms among Group GP and Group GP + CHD in the subgingival plaque and blood samples.

IL-1β (+3954) Subgingival Plaque	Group GP	Group GP + CHD	Odds Ratio^b^ ^(Class^ ^Interval)^	*p*-Value^c^
n	27 (29)	32		
Frequency %	93.10	100		
CT	14	9	1.14 (1.04–1.19)	0.052^NS^
AT	3	10	0.92(0.88–1.09)	0.01*
AG	10	13	0.91 (0.87–1.07)	0.032*
**IL-1β (+3954) Blood**	**Group GP**	**Group GP + CHD**	**Odds Ratio^b^** **^(Class Interval)^**	***p*-value^c^**
N	25	29 (35)		
Frequency %	100	82.85		
Allele change (Frequency)^a^ n(%)				
CT	10	7	1.28 (0.95–1.32)	0.026*
AT	8	13	0.97 (0.87–1.20)	0.007^†^
AG	7	9	0.88 (0.82–0.96)	0.78 ^NS^

* *p* < 0.05 is considered to be statistically significant. † *p* < 0.01 is considered to be highly statistically significant. NS—Nonsignificant. ^a^ Values are given as n (%) of individuals. ^b^ Odds ratio (95% confidence interval). ^c^ Two-sided Pearson’s Chi-square test (X^2^). A—Adenine. C—Cytosine. G—Guanine. T—Thymine.

**Table 4 materials-14-05103-t004:** Overall comparison of the demographic variables, periodontal parameters, and cardiac parameters with the allele change (frequency) of IL-1β (+3954) in the subgingival plaque and blood samples for both the groups.

Variable	Allele Change (Frequency) Mean ± Standard Deviation (Subgingival Plaque Sample)			*p*-Value	Allele Change (Frequency) Mean ± Standard Deviation (Blood Sample)			
	CT	AT	AG		CT	AT	AG	*p* Value
**Age (years)**	50.6 ± 11.2	56.2 ± 13.3	38 ± 12.4	0.001^‡^	46.6 ± 7.8	52.2 ± 9.9	34 ± 9	0.05^*^
**Height (cm)**	168.7 ± 11	174.2 ± 16.1	172.2 ± 9.1	0.867 ^NS^	164.7 ± 7.8	163.2 ± 6.7	167 ± 5.7	0.379 ^NS^
**Weight (Kg)**	66.6 ± 6.1	74.8 ± 20.4	67.8 ± 8.9	0.513 ^NS^	60.6 ± 2.5	68.8 ± 9	63.8 ± 3.5	0.714 ^NS^
**Monthly Income (Rs)**	25,285.71 + 7204.496	22,600 + 7916.7	28,500 + 8888.2	0.11 ^NS^	20,769.23 + 6002.136	31,666.67 + 5773.503	20,076.92 + 7750.930	0.01^†^
**Body Mass Index (BMI)**	26.5 ± 7.3	32.2 ± 10.6	30.4 ± 4.8	0.262 ^NS^	22.5 ± 1.9	25 ± 3.2	23 ± 1.4	0.368 ^NS^
**Waist–Hip ratio**	2.8 ± 3.4	4.9 ± 3.4	4.8 ± 3.4	0.447 ^NS^	0.8 ± 0.03	0.9 ± 0.04	0.8 ± 0.03	0.952 ^NS^
**Total Cholesterol (mg/dl)**	176.4 ± 32.8	180.6 ± 30.4	178.4 ± 71.1	0.114 ^NS^	159.3 ± 6.3	181.4 ± 27	194.3 ± 33.4	0.867 ^NS^
**High-Density Lipoprotein (mg/dl)**	37.6 ± 8.7	38.5 ± 7.2	39.3 ± 9.3	0.437 ^NS^	39.8 ± 6.6	35.1 ± 3.2	36 ± 9.6	0.513 ^NS^
**Low-Density Lipoprotein (mg/dl)**	115.7 ± 33.6	119.9 ± 33.3	118.5 ± 45.6	0.321 ^NS^	97.5 ± 9.3	122.8 ± 20.7	134.3 ± 33.7	0.11 ^NS^
**Triglycerides (mg/dl)**	144.7 ± 60	124.4 ± 40.5	130.1 ± 29.9	0.197 ^NS^	141.5 ± 18.9	120.8 ± 32.4	171.2 ± 66.8	0.262 ^NS^
**Systolic Blood Pressure (mm Hg)**	124.3 ± 5.3	129 ± 12.1	117.5 ± 5	0.1 ^NS^	125 ± 5.8	124.3 ± 11.3	128 ± 10.3	0.522 ^NS^
**Diastolic Blood Pressure (mm Hg)**	74.3 ± 5.3	80.5 ± 5.1	75 ± 5.8	0.268 ^NS^	72.5 ± 5	79.3 ± 4.9	82 ± 6.3	0.1 ^NS^
**Plaque Index**	0.7 ± 0.1	0.8 ± 0.3	0.7 ± 0.1	0.1 ^NS^	0.7 ± 0.1	0.9 ± 0.1	1 ± 0.2	0.05^*^
**Bleeding on Probing (%)**	74.4 ± 2.9	71.4 ± 9.6	75.1 ± 27	0.1 ^NS^	74.5 ± 18.6	68.9 ± 19.3	77.2 ± 25.5	0.001^‡^
**Probing Pocket Depth (mm)**	4.3 ± 0.4	3.7 ± 0.9	4.6 ± 0.3	0.67^NS^	3.9 ± 0.2	3.9 ± 0.9	4.2 ± 0.3	0.715 ^NS^
**Clinical Attachment Level (mm)**	5.3 ± 0.6	4.6 ± 0.9	5.7 ± 0.8	0.01^†^	4.7 ± 0.3	5 ± 1	4.6 ± 0.2	0.001^‡^

* *p* < 0.05 is considered to be statistically significant. † *p* < 0.01 is considered to be highly statistically significant. ‡ *p* < 0.001 is considered to be very highly statistically significant. NS—Nonsignificant.

## Data Availability

All the data is available within the manuscript.

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
