# Peer review of "The Expression of Allele Changes in NLRP3 (rs35829419) and IL-1β (+3954) Gene Polymorphisms in Periodontitis and Coronary Artery Disease"

_materials, 2021, doi:10.3390/ma14175103_

Round 1
Reviewer 1 Report
The manuscript is devoted to the study of the relationship between the NLRP3 (rs35829419) and IL-1β (+3954) SNPs and the etiopathogenesis of periodyntitis separately and in combination with coronary artery disease.
There are a number of comments on the text of the article:
- There is no information in the introduction - why the authors decided to study a group with a combination of CP and CAD. What research has been done on this issue earlier? What is already known on this issue that remains unclear? This section needs to be completed.
- It remains unclear from the study design - how it was possible to form a cohort of patients with such an accurate division into groups (35 in each group). Apparently, some kind of special selection was carried out, this must be described.
- I would also like to clarify the criteria for establishing the diagnosis of coronary artery disease by a cardiologist (clinical picture, data from non-invasive tests, coronary angiography, myocardial infarction)?
- The style of the Discussion section also requires correction. This section provides extensive information on the results obtained by the authors, even Table 4. These data should be presented in the Results section.
- Since the CP and CP + CAD groups differed in age, it remains unclear to what extent the changes in the allele of the IL-1β gene polymorphism (+3954) are due to the additional detection of CAD, rather than age-related changes. Additional statistical analysis appears to be needed to answer this question.
- The article does not contain a Study limitation section; you need to add it.
Author Response
Reviewer 1:
The manuscript is devoted to the study of the relationship between the NLRP3 (rs35829419) and IL-1β (+3954) SNPs and the etiopathogenesis of periodontitis separately and in combination with coronary artery disease.
There are a number of comments on the text of the article:
- There is no information in the introduction - why the authors decided to study a group with a combination of CP and CAD. What research has been done on this issue earlier? What is already known on this issue that remains unclear? This section needs to be completed.
Ans: Appropriate changes have been made in the Introduction section.
- It remains unclear from the study design - how it was possible to form a cohort of patients with such an accurate division into groups (35 in each group). Apparently, some kind of special selection was carried out, this must be described.
Ans: Cohorts were selected and divided based on the criteria for periodontal disease as given by World workshop in Classification of Periodontal Diseases and Conditions 1999 and for group II, patients were selected based on cardiologist diagnosis. Till the power of the study was reached, the study participants were obtained for both the groups.
- I would also like to clarify the criteria for establishing the diagnosis of coronary artery disease by a cardiologist (clinical picture, data from non-invasive tests, coronary angiography, myocardial infarction)?
Ans: ). For both, CP and CP+CHD group , the patients for cardiac problems were diagnosed based on clinical examination, weather patient is symptomatic or asymptomatic, ECG, blood test for Lipid profile and LDH followed by echocardiogram. Based on the above diagnostic parameters, the appropriate diagnosis was made by the cardiologist for the patients.
Necessary changes has been made in the manuscript.
- The style of the Discussion section also requires correction. This section provides extensive information on the results obtained by the authors, even Table 4. These data should be presented in the Results section.
Ans: Discussion has been modified according to reviewer’s comments and Table 4 has been elaborated in the discussion section with appropriate references.
- Since the CP and CP + CAD groups differed in age, it remains unclear to what extent the changes in the allele of the IL-1β gene polymorphism (+3954) are due to the additional detection of CAD, rather than age-related changes. Additional statistical analysis appears to be needed to answer this question.
Ans: As both periodontal disease and CHD are age related diseases and both are mostly prevalent above 40 years of age, hence the difference between the CP and CP+CHD groups with respect age would hardly make a difference when compared as both the patients fell into the same age range of 40-60 years.
- The article does not contain a Study limitation section; you need to add it.
Ans: Limitation section has been added according to reviewer’s response

Reviewer 2 Report
It is undoubtedly an interesting and valuable article. As the authors emphasize, this study aimed to obtain the expression of NLRP3 (rs35829419) and IL-1β (+3954) gene polymorphisms in subgingival plaque and blood samples of generalized chronic periodontitis (CP) subjects with and without coronary heart disease (CHD). The general comparison of all the parameters with the allele change of NLRP3 (rs35829419) and IL-1β (+3954) in subgingival plaque and blood samples showed statistically significant associations among the two groups selected from the patients enrolled in these studies. The authors rightly concluded that their study expressed an allele change of IL-1β (+3954) gene polymorphisms which may play an important role in the pathogenesis of periodontitis and coronary heart disease. Undoubtedly, the work is part of the trend of those that confirm that there is a cause-and-effect relationship between oral diseases and systemic diseases. Hence, treatment of all oral diseases, including periodontal diseases, is necessary to ensure good general health and well-being of people.
Except that this work is not even one percent of the scope of the Materials journal, it does not respond to the call of this Special Issue on "Advances in Periodontics and Restorative Dental Materials".
It is imperative that every article in Materials meets the six-expectation of the Materials Science paradigm, known as the 6xE Principle. In general, the point here is to find relationships between the causes of oral diseases and possible engineering materials and material process technologies that can best help directly, but even indirectly, in dental therapy. In this case, not even such an attempt was made, and moreover, not a single sentence was devoted to this aspect/context. Even an excuse was not made to indicate why this paper was submitted to Materials.
The work does not cover any engineering material aspects and should therefore be strongly rejected in Materials. In this journal, the issues presented in their entirety in this article could at most be a fragment of an introduction to an interdisciplinary study on dentistry/medicine/microbiology and engineering materials used in diagnostics/therapy. The authors did not construct their study in this way.
The work should be strongly rejected and sent to a medical or microbiology journal.
Author Response
Reviewer 2:
It is undoubtedly an interesting and valuable article. As the authors emphasize, this study aimed to obtain the expression of NLRP3 (rs35829419) and IL-1β (+3954) gene polymorphisms in subgingival plaque and blood samples of generalized chronic periodontitis (CP) subjects with and without coronary heart disease (CHD). The general comparison of all the parameters with the allele change of NLRP3 (rs35829419) and IL-1β (+3954) in subgingival plaque and blood samples showed statistically significant associations among the two groups selected from the patients enrolled in these studies. The authors rightly concluded that their study expressed an allele change of IL-1β (+3954) gene polymorphisms which may play an important role in the pathogenesis of periodontitis and coronary heart disease. Undoubtedly, the work is part of the trend of those that confirm that there is a cause-and-effect relationship between oral diseases and systemic diseases. Hence, treatment of all oral diseases, including periodontal diseases, is necessary to ensure good general health and well-being of people.
Except that this work is not even one percent of the scope of the Materials journal, it does not respond to the call of this Special Issue on "Advances in Periodontics and Restorative Dental Materials".
It is imperative that every article in Materials meets the six-expectation of the Materials Science paradigm, known as the 6xE Principle. In general, the point here is to find relationships between the causes of oral diseases and possible engineering materials and material process technologies that can best help directly, but even indirectly, in dental therapy. In this case, not even such an attempt was made, and moreover, not a single sentence was devoted to this aspect/context. Even an excuse was not made to indicate why this paper was submitted to Materials.
The work does not cover any engineering material aspects and should therefore be strongly rejected in Materials. In this journal, the issues presented in their entirety in this article could at most be a fragment of an introduction to an interdisciplinary study on dentistry/medicine/microbiology and engineering materials used in diagnostics/therapy. The authors did not construct their study in this way.
The work should be strongly rejected and sent to a medical or microbiology journal.
Ans: Respected reviewer, we beg to bring to your notice that this article was submitted with the prime intent to highlight the advances in periodontics and since this journal had an upcoming special issue in advances in periodontics and restorative dentistry we submitted the present article. We request you to kindly consider the same for publication as it is eligible in this regard

Reviewer 3 Report
The article cannot be accepted in this journal because it covers a topic not included in the journal's interests (class of materials, materials engineering, nanoscience and nanotechnology ...). I recommend looking for an alternative journal to publish these results.
In any case, the results shown are descriptive and not very clarifying. The authors must be more ambitious and try to explain the biological consequences of the genetic polymorphisms studied. The plasma levels of IL-1beta should be evaluated in the two groups of patients to try to correlate the genotypes with the degree of the inflammasome activation.
Author Response
Reviewer 3:
The article cannot be accepted in this journal because it covers a topic not included in the journal's interests (class of materials, materials engineering, nanoscience and nanotechnology ...). I recommend looking for an alternative journal to publish these results.
In any case, the results shown are descriptive and not very clarifying. The authors must be more ambitious and try to explain the biological consequences of the genetic polymorphisms studied. The plasma levels of IL-1beta should be evaluated in the two groups of patients to try to correlate the genotypes with the degree of the inflammasome activation.
Ans: Respected reviewer, we beg to bring to your notice that this article was submitted with the prime intent to highlight the advances in periodontics and since this journal had an upcoming special issue in advances in periodontics and restorative dentistry we submitted the present article. We request you to kindly consider the same for publication as it is eligible in this regard
The reviewer’s suggestion has been also been well taken. Biologic consequences of genetic polymorphisms have been elaborated and our future studies will be aimed at comparing the degree of inflammasome activation.

Reviewer 4 Report
It requires minor revision before possible publication in Materials.
The paper is well written and planned. I would consider possibly transferring it to JCM or Genes, whose aims and scopes would be more in accordance with the subject undertaken.
The paper is easy to read.
1.INTRODUCTION: Please explain the purpose of the paper in more detail, why the authors were interested in this topic, and why they posed such a research question. Would you please provide a rationale for undertaking such a topic?
2. MATERIALS AND METHODS: Please explain the inclusion and exclusion criteria of the study in more detail.
3. RESULTS: The results are clearly presented
4. Discussion: The discussion should be changed in some places. Would you please focus more on relating the results of the paper to clinical situations and what future benefits the results can give us?
5. Bibliography. The number of references should be before the full stop.
6. Limitations of the study are missing.
7. CONCLUSION: they address the main question.
Author Response
Reviewer 4:
It requires minor revision before possible publication in Materials.
The paper is well written and planned. I would consider possibly transferring it to JCM or Genes, whose aims and scopes would be more in accordance with the subject undertaken.
The paper is easy to read.
1.INTRODUCTION: Please explain the purpose of the paper in more detail, why the authors were interested in this topic, and why they posed such a research question. Would you please provide a rationale for undertaking such a topic?
Ans: Appropriate changes have been made in the Introduction section.
MATERIALS AND METHODS: Please explain the inclusion and exclusion criteria of the study in more detail.
Ans: the inclusion and exclusion criteria has been explained and highlighted as per the reviewer’s suggestion.
- RESULTS: The results are clearly presented
- Discussion: The discussion should be changed in some places. Would you please focus more on relating the results of the paper to clinical situations and what future benefits the results can give us?
Ans: Discussion has been modified in few places and the clinical implication has been elaborated according to reviewer’s comments.
Bibliography. The number of references should be before the full stop.
Ans: the reference has been modified according to reviewer’s comments.
- Limitations of the study are missing.
Ans: Limitation section has been added according to reviewer’s response
CONCLUSION: they address the main question.

Round 2
Reviewer 1 Report
After revision, the manuscript has been greatly improved and can be published in Materials.
Author Response
Reviewer 1:
After revision, the manuscript has been greatly improved and can be published in Materials.
Response: respected reviewer, we humbly thank you for your appreciation and recommendation for acceptance
Reviewer 2:
Scope of the Materials magazine we find information about Topics Covered as follows:
Class of materials include ceramics, glasses, polymers (plastics), semiconductors, magnetic materials, medical implant materials and biological materials, silica and carbon materials, metals and metallic alloys. All kinds of functional materials including material for dentistry. Composites, coatings and films, pigments. Classes of materials such as ionic crystals; covalent crystals; metals; intermetallics.
Materials science or materials engineering. Nanoscience and nanotechnology will be considered also.
Characterization techniques such as electron microscopy, x-ray diffraction, calorimetry, nuclear microscopy (HEFIB), Rutherford backscattering, neutron diffraction, etc.
Fundamental research: Condensed matter physics and materials physics. Continuum mechanics and statistics. Mechanics of materials. Tribology (friction, lubrication and wear). Solid-state physics.
In turn, these issues are narrowed down to the issues of materials used in periodontics and materials used for restorative dentistry, which is as follows:
The Special Issue on “Advances in Periodontics and Restorative Dental Materials” aims to bring together articles covering relevant scientific topics on periodontal disease diagnosis, pathogenesis, and advances in therapy. Moreover, the opportunity will be taken to discuss the state-of-the-art research on materials that could be used in restorative dentistry and endodontics with a positive influence on periodontal conditions. To this end, the methods, requested devices, and processing of materials could be high-tech and key to developing high-quality competitive products also going beyond these specific topics. This Special Issue will consider the foundation of technical progress in periodontology and dentistry, the development of novel diagnostic and therapeutic possibilities, and the production of novel materials in creating new designs for instruments and precision machinery as well as in developing technologies and systems.
This Special Issue is a timely approach to survey recent progress in the development and optimization of these subjects. The articles presented in this Special Issue will cover various topics ranging from advances in periodontal diagnosis, pathogenesis, therapy, endoperiodontal treatment, direct and indirect dental restoration, and studies on oral microbiota. Therefore, this Special Issue welcomes contributions from all researchers working in omics, biomolecular sciences, or materials processing in relation to periodontal health and dental restoration.
There is no doubt that gene therapies and similar therapies do not fit into the given topic.
It is not a magazine for people who are not interested in the materials. The title of the magazine itself indicates the target audience.
Each self-respecting journal cannot be a conglomerate of articles on all possible topics that anyone will associate with any word taken out of context. On the one hand, it is a break of respect for the intelligence of readers and editors, and on the other hand, it is a huge arrogance of the authors not to react properly to the kind remark regarding the inappropriate submission of an article to a journal, the scope of which does not concern the issues discussed in this journal. It is undoubtedly also a violation of the ethics of researchers, as well as a violation of good practice in science. Completely incomprehensible in this context is the authors' response to the reviewer's previous comments as follows:
"we beg to bring to your notice that this article was submitted with the prime intent to highlight the advances in periodontics".
Of course, it is about materials related to periodontics, and the article is not about it.
And further as follows: "this journal had an upcoming special issue in advances in periodontics". This is simply not true, as it is all about advances in periodontics in the context of the materials that are used there.
The authors' statement "it is eligible in this regard" is also untrue, because in the paper there is absolutely no question of materials for periodontology, but it is quite the opposite.
Of course, the editors of Special Issue should be aware of what they expect from the authors, and on the other hand, they should take responsibility for the shape and level of the journal and not make it an outlet for any topics that, for various reasons, are not published in their thematically relevant journals.
I am definitely against this article because it definitely does not fit into the scope of the Materials journal and does not fit into the scope of the special issue, which should apply to materials for periodontology, even if we approach this topic strongly and broadly.
The paper should be rejected.
Response: Respected reviewer: we humbly request you to consider our efforts . I wish to humbly place on record that genetic research is an advance in periodontology and we have submitted a novel article in this regard. We submitted the same to your journal considering the fact that it is appropriate for the special issue. Please help us in this regard.
Reviewer 3:
The authors have not addressed my concerns and their explanations are not convincing.
Academic editor comments:
The study considers the relationship between the NLRP3 (rs35829419) and IL-1β (+3954) SNPs and the etiopathogenesis of periodontitis and coronary artery disease. It is an interesting field of study. However, it is necessary to make several parts clear.
Q1 - It is necessary to accurately describe what the recorded periodontal indices correspond to. Is the PPD reported to be considered in relation to the periodontal site (s) where the periodontal plaque sample(s) is (are) collected or is it the average of the all PPD measured on all dental sites? In this case, how many measurements were carried out per tooth (4 .... 6)?
Ans: The probing pocket depth is an indicator of past periodontal destruction and in the current study PPD was measured at 6 sites per tooth in all the teeth of the dentition and was averaged.
Q2: The same matter goes for the CAL. Moreover, CAL is quite often considered as the sum of PPD + REC. Was it so or was it measured directly?
Ans: The CAL denotes the past periodontal destruction and tissue loss. CAL also was measured at 6 sites per tooth of all teeth in the dentition and was averaged. CAL was measured directly.
Q3: Was PI detected at dental plaque sample collection sites or is it similar to full-mouth bleeding scores / full-mouth plaque scores and quoted as percentage? Were BOP and PI measured at 2, 4, 6 sites per tooth?
Ans: The plaque index (PI) of the current study was recorded according to the criteria of Silness and Loe (1964) (12) in 4 sites all teeth of the dentition (full mouth plaque scores). BOP and PI were both measured in 4 sites all teeth of the dentition. The reference for the measurement of PI has also been quoted in the manuscript.
Q4: Authors should also specify how PI was assessed.
Ans: The plaque index score for the area was obtained by totalling the four plaque scores per tooth. The sum of the plaque index scores per tooth was divided by four to obtain the plaque index score for the tooth. The same has been included in the manuscript.
Q5: Also BP recording could be quite different. What were the criteria? Were both the ambulatory pressure monitoring (ABPM) and home blood pressure monitoring (HBPM) measured (Whelton et al. 2018)?
Ans: The blood pressure recordings of the patients were taken from the medical records of the hospital when patient was at rest. In our study for both the groups, blood pressure was recorded when patient was at a relaxed state.
Q6: How were the other indices measured? For example, were height and weight measured or reported by patients?
Ans: Height, weight, BMI were all recorded from hospital records for patients in Group GP+CHD patients. Group GP patients were advised to undergo height, weight and waist-hip ratio measurements. BMI was calculated based on height and weight recordings.
Q7: The discussion is verbose. It has to be rationalized and made immediate, hinging it on the meaning of the study.
Ans: Reviewer’s comments are well taken. Discussion has been modified according to reviewer’s suggestions and has been appropriately incorporated in the manuscript.
Q8: The study was performed during the period December 2018-2020. The latest classification of periodontal diseases (consensus report of workgroups of the 2017) was published within june 2018. Why did the Authors follow the 1999 classification?
Ans: As per the reviewer’s suggestion, the case definition of periodontitis patients in GP and GP+CHD groups has been modified in the manuscript appropriately according to the 2017 World Workshop of Periodontics Classification of periodontitis and peri-implant diseases.

Reviewer 2 Report
Scope of the Materials magazine we find information about Topics Covered as follows:
Class of materials include ceramics, glasses, polymers (plastics), semiconductors, magnetic materials, medical implant materials and biological materials, silica and carbon materials, metals and metallic alloys. All kinds of functional materials including material for dentistry. Composites, coatings and films, pigments. Classes of materials such as ionic crystals; covalent crystals; metals; intermetallics.
Materials science or materials engineering. Nanoscience and nanotechnology will be considered also.
Characterization techniques such as electron microscopy, x-ray diffraction, calorimetry, nuclear microscopy (HEFIB), Rutherford backscattering, neutron diffraction, etc.
Fundamental research: Condensed matter physics and materials physics. Continuum mechanics and statistics. Mechanics of materials. Tribology (friction, lubrication and wear). Solid-state physics.
In turn, these issues are narrowed down to the issues of materials used in periodontics and materials used for restorative dentistry, which is as follows:
The Special Issue on “Advances in Periodontics and Restorative Dental Materials” aims to bring together articles covering relevant scientific topics on periodontal disease diagnosis, pathogenesis, and advances in therapy. Moreover, the opportunity will be taken to discuss the state-of-the-art research on materials that could be used in restorative dentistry and endodontics with a positive influence on periodontal conditions. To this end, the methods, requested devices, and processing of materials could be high-tech and key to developing high-quality competitive products also going beyond these specific topics. This Special Issue will consider the foundation of technical progress in periodontology and dentistry, the development of novel diagnostic and therapeutic possibilities, and the production of novel materials in creating new designs for instruments and precision machinery as well as in developing technologies and systems.
This Special Issue is a timely approach to survey recent progress in the development and optimization of these subjects. The articles presented in this Special Issue will cover various topics ranging from advances in periodontal diagnosis, pathogenesis, therapy, endoperiodontal treatment, direct and indirect dental restoration, and studies on oral microbiota. Therefore, this Special Issue welcomes contributions from all researchers working in omics, biomolecular sciences, or materials processing in relation to periodontal health and dental restoration.
There is no doubt that gene therapies and similar therapies do not fit into the given topic.
It is not a magazine for people who are not interested in the materials. The title of the magazine itself indicates the target audience.
Each self-respecting journal cannot be a conglomerate of articles on all possible topics that anyone will associate with any word taken out of context. On the one hand, it is a break of respect for the intelligence of readers and editors, and on the other hand, it is a huge arrogance of the authors not to react properly to the kind remark regarding the inappropriate submission of an article to a journal, the scope of which does not concern the issues discussed in this journal. It is undoubtedly also a violation of the ethics of researchers, as well as a violation of good practice in science. Completely incomprehensible in this context is the authors' response to the reviewer's previous comments as follows:
"we beg to bring to your notice that this article was submitted with the prime intent to highlight the advances in periodontics".
Of course, it is about materials related to periodontics, and the article is not about it.
And further as follows: "this journal had an upcoming special issue in advances in periodontics". This is simply not true, as it is all about advances in periodontics in the context of the materials that are used there.
The authors' statement "it is eligible in this regard" is also untrue, because in the paper there is absolutely no question of materials for periodontology, but it is quite the opposite.
Of course, the editors of Special Issue should be aware of what they expect from the authors, and on the other hand, they should take responsibility for the shape and level of the journal and not make it an outlet for any topics that, for various reasons, are not published in their thematically relevant journals.
I am definitely against this article because it definitely does not fit into the scope of the Materials journal and does not fit into the scope of the special issue, which should apply to materials for periodontology, even if we approach this topic strongly and broadly.
The paper should be rejected.
Author Response

(The authors gave the same response as above.)

Reviewer 3 Report
The authors have not addressed my concerns and their explanations are not convincing.
Author Response

(The authors gave the same response as above.)
